# Exploring the Effect of Pt Addition on the Fracture Behavior of CrN Coatings by Finite Element Simulation

Haifeng Sun [1,†], Weilun Zhang [1,†], Yongjun Feng [1,*], Suying Hu [1], Hua Tian [2,*] and Zhiwen Xie [1]

1 Liaoning Key Laboratory of Complex Workpiece Surface Special Machining, University of Science and Technology Liaoning, Anshan 114051, China
2 Equipment Manufacturing Department, Anshan Vocational and Technical College, Anshan 114020, China
* Correspondence: xfsword@163.com (Y.F.); 13804129210@163.com (H.T.)
† The authors contributed equally to this work.

**Abstract:** Previous research confirmed that Pt addition induced a prominent refinement effect of CrN coating, resulting in an enhanced conductivity and corrosion resistance. In this work, a detailed finite element simulation and scratch test were employed to calculate and characterize the fracture failure behaviors (stress distribution, crack damage process, critical coating load, and coating–substrate adhesion energy) of CrN coatings with different Pt contents. Simulation results showed that the synergistic action of dynamic scratch load and extrusion load induced the fracture of the coatings. S11 and S22 caused transverse cracks in the CrN coating, S11 caused longitudinal cracks in the CrN-Pt coating and CrN-3Pt coatings, S22 led to the inclined propagation of cracks in these coatings, and S11 and S22 jointly induced the separation of the coating from the substrate. The doping Pt element in the CrN coating will make the coating easier to fracture and reduce the adhesion strength between the coating and substrate. Scratch test results revealed that adding Pt into the CrN coating will make this coating easier to fracture and cause more serious damage; the simulation results are in good accordance with the scratch test characterizations. The current founding provided a comprehensive understanding for the fracture damage mechanism of Pt-doped nitride coatings.

**Keywords:** CrN coating; Pt elements; fracture behavior; adhesion strength; finite element simulation

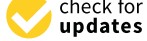



## 1. Introduction

When electrical connectors are used in a marine environment, oxidation reactions will occur, which aggravate electrochemical corrosion and reduce the efficiency of signal transmission. To enhance the reliability of electrical connections in the marine environment, we used CrN coatings doped with Pt elements to protect their surfaces. CrN coating has good adhesion to the substrate, a low friction coefficient and good oxidation resistance. It also has good wear resistance and corrosion resistance in harsh environments such as high temperature and high pressure [1–4]. It is widely used as a protective coating for tensile dies, cutting tools and various moving parts in machinery and equipment [5–8]. However, the high temperature thermal stability of CrN coating is poor, and the hardness and corrosion resistance of the coating will be reduced in the marine environment with high salinity and humidity [9]. Studies have shown that adding elements such as W, Al, Ti, C to CrN coatings can improve the mechanical properties and corrosion resistance of the coatings [10–13]. Pt elements have the advantages of high melting point, high stability, high strength and high hardness [14,15]. The research found that there were few research results reported on CrN coatings doped with Pt elements. Therefore, we prepared CrN (Pt doping 0), CrN-Pt (Pt doping 5.47 at %) and CrN-3Pt (Pt doping 15.00 at %) coated specimens using 316L stainless steel as the substrate, and we carried out a systematic study on the service conditions of electrical connections in the marine environment using a plasma-enhanced magnetron sputtering process.

Previous research found that doping Pt elements in CrN coating can enhance the refinement and densification effect of microstructure, improve its conductivity and corrosion resistance, and increase positively with the increase in Pt content [16]. Extended finite element method (XFEM) is mainly used to solve the problems of crack initiation and propagation in linear elasticity. It is the most effective numerical simulation method to solve discontinuous problems [17]. In order to improve the working stability of the coating under dynamic friction and extrusion load, this paper uses the XFEM method to explore the influence of doping different contents of Pt elements on the fracture behavior of CrN coating. At the same time, the results of scratch test and finite element simulation are compared and analyzed to verify the accuracy of the finite element results, which provides support for the design and preparation of CrN protective coating for electrical connectors.

## 2. Material and Methods

### 2.1. Material

We ultrasonically cleaned and polished the 316L stainless steel substrate with a size of Φ 25 mm × 8 mm. The elemental composition of the 316L stainless steel substrate is shown in Table 1. CrN coating, CrN-Pt coating and CrN-3Pt coating were prepared on the 316L stainless steel substrate by plasma-enhanced magnetron sputtering technology. The elemental compositions of the three coatings are shown in Table 2. The cross-sectional structure was observed by SEM, and the thicknesses of the three coatings were measured as shown in Table 2.

**Table 1.** Matrix element composition of 316L stainless steel (wt.%).

| Fe | Cr | Ni | C | Mo | Mn | Si |
|---|---|---|---|---|---|---|
| 66.63 | 15.66 | 11.56 | 1.98 | 1.87 | 1.84 | 0.46 |

**Table 2.** Coating parameters.

| Coating | Thickness (μm) | Element (at %) | | |
|---|---|---|---|---|
| | | Cr | N | Pt |
| CrN | 6.9 | 52.01 | 47.99 | 0 |
| CrN-Pt | 7.03 | 50.13 | 44.40 | 5.47 |
| CrN-3Pt | 10.35 | 48.26 | 36.74 | 15.00 |

### 2.2. Finite Element Simulation Model

Geometric model: Extended finite element method (XFEM) was used to simulate the coating nanoscratch process [18–21]. Three two-dimensional axisymmetric simulation models were created (take the right half, see Figure 1). The substrate width (X-direction, transverse) is 120 μm, the thickness (Y-direction, longitudinal) is 50 μm, and the thicknesses of the CrN, CrN-Pt, and CrN-3Pt coatings are shown in Table 2. A conical rigid indenter with a taper angle of 65° and a radius of 200 nm is used, and the elastic modulus and Poisson's ratio of the indenter are shown in Table 3. Only the stresses in the XY plane were analyzed, and S11 was defined as the stress component in the X-direction, S22 was defined as the stress component in the Y-direction, and S12 was defined as the shear stress component; the tensile stress was a positive value and the compressive stress was a negative value.

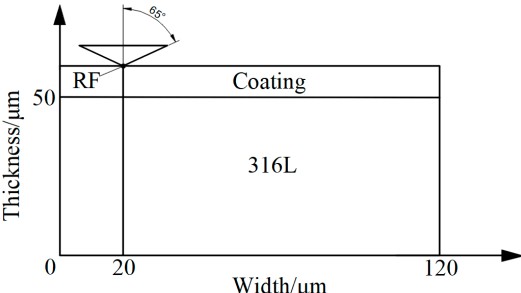

**Figure 1.** Schematic diagram of the overall assembly of the two-dimensional model.

**Table 3.** Parameters of the finite element simulation model of the scratch process.

| Material | Maximum Principal Stress GPa | Fracture Energy MPa × m$^{1/2}$ | Yield Strength GPa | Tangent Modulus GPa | Elastic Modulus GPa | Poisson Ratio | Friction Coefficient |
|---|---|---|---|---|---|---|---|
| CrN | 0.43 | 17.26 | 0.41 | 33.44 | 247.7 | 0.29 | 0.74 |
| CrN-Pt | 0.65 | 16.84 | 0.63 | 26.30 | 284.6 | 0.29 | 0.83 |
| CrN-3Pt | 0.95 | 16.52 | 0.93 | 24.27 | 261.5 | 0.29 | 0.77 |
| 316L | 0.93 | 13.25 | 0.91 | 28.15 | 135 | 0.30 | — |
| Indenter | — | — | — | — | 1141 | 0.07 | — |

Model parameters: During the scratching process, both the coating and the substrate undergo elastic deformation and plastic deformation. The elastic modulus of the three coatings were obtained through the nanoindentation test, the finite element simulation of the nanoindentation test was performed to obtain the plastic parameters, and we also calculated the fracture energy of the three coatings [16]. All parameters of the finite element simulation model of the scratching process are shown in Table 3.

Constraints: Nodes on the left side of the model can only be displaced in the Y-direction, and the nodes on the bottom side of the model can only be displaced in the X-direction. The right side and top of the model are not constrained.

Mesh division: A four-node symmetric linear reduced integral cell (CAX4R) was used for meshing; the mesh properties were quadrilateral structural mesh, and the mesh of the crack emergence region (coating part) was encrypted. The finer the mesh division, the more conducive to the initiation and propagation of cracks [22,23].

Load application: At the beginning of the simulation, the normal load (Y-direction) is applied to the reference point RF of the indenter to simulate the extrusion load during the connection of electrical connectors. At the same time, the indenter is made to slide 100 μm uniformly from 20 μm to the right along the transverse direction (X-direction); to simulate the dynamic friction load during the plugging and unplugging of electrical connectors, no cracks are preset. There is sliding friction between the indenter and the coating surface, and the friction coefficient is obtained by the friction test. Friction coefficients of the three coatings are shown in Table 3. Normal load is set to 60, 80 and 100 mN, respectively, and the final load state where the coating produces penetration cracks and film base separation occurs is taken for analysis.

### 2.3. Study on Fracture Behavior

The fracture resistance of the coating is closely related to the residual stress on the surface of the coating. After the scratch process is completed, the isotropic residual stress on the surface of the three coatings is extracted, and the dominant residual stress component is determined. The maximum tensile stress value of the dominant residual stress component is extracted, and the fracture behavior of the three coatings is compared, analyzed, and evaluated [24].

The adhesion energy of the coating–substrate of the three coatings was calculated to compare and evaluate the bonding strength of the coating–substrate of the coatings. Combining the equation of film-based binding energy $W$ versus interfacial stress $\sigma$ during

scratching of hard coatings given by Laugier [25] and the equation of equivalent force by Attar and Bull [26], the equation of the adhesion energy of the coating–substrate versus critical load can be obtained (1).

$$W = \frac{d}{2E_f} \left[ \frac{v_f \mu_c L_c}{da_c} \right]^2 \qquad (1)$$

In the formula: $W$ is the binding energy of the film base; $d$ is the thickness of the coating; $E_f$ is the elastic modulus of the coating; $v_f$ is Poisson's ratio; $\mu_c$ is the friction coefficient; and $L_c$ is the critical load. The normal load applied by the indenter when the crack completely penetrates and extends to the coating and substrate interface is the critical load; $a_c$ is the corresponding width of the scratch at the location where the critical load occurs.

### 2.4. Scratch Test

The coating scratch test was used to simulate the working conditions of electrical connections protected by CrN coating, and the protection effects of the three coatings were compared and analyzed by observing the microscopic morphology of the scratches after the scratch test [27–30]. We scratch tested the coating with a scratch tester (HST-200, AiRTX, San Francisco, CA, USA). The loading force is 0–100 N, the loading speed is 40 N/min, and the scratch length is 5.00 mm. We used the ultra-depth of field equipment (KEYENCE, VHX-5000, Osaka, Japan) to observe the coating morphology after the scratch test.

## 3. Results

### 3.1. Analysis of Stress Distribution and Fracture Behavior of Coating

The comparative simulation analysis reveals that the normal load of 100 mN produces penetration cracks and separation of the coating–substrate for all three coatings, so such load simulation results are taken for analysis. Figure 2a–c shows the stress nephogram of CrN, CrN-Pt and CrN-3Pt coatings when the crack extends to the coating–substrate interface under a normal load of 100 mN.

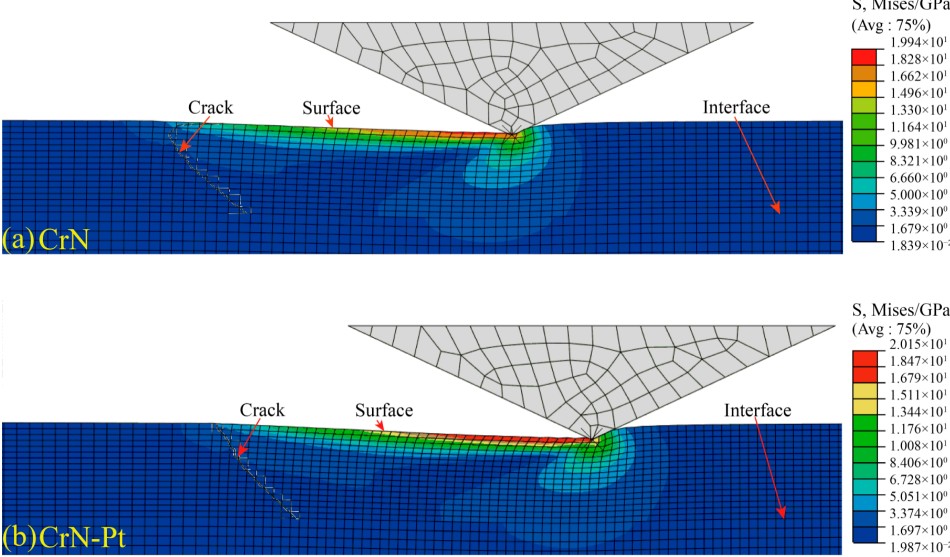

**Figure 2.** *Cont.*

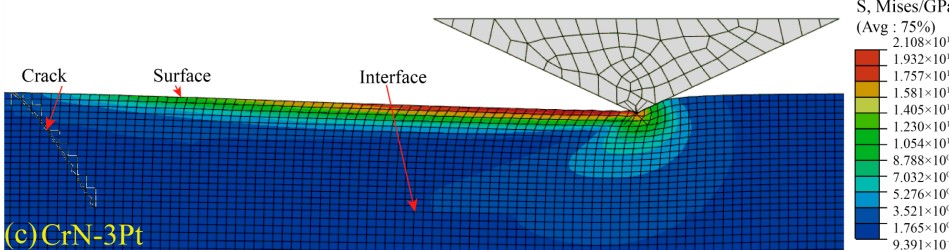

**Figure 2.** Stress nephogram of three kinds of coating cracks when penetrating the interface between coating and substrate (normal load is 100 mN). (**a**) CrN; (**b**) CrN-Pt; (**c**) CrN-3Pt.

An analysis of Figure 2 shows that the indenter applies Y-direction pressure and X-direction friction to the coating during the sliding process on the coating surface, forming stresses on the coating surface and inside. When the stress on the surface of the coating is greater than the yield strength of the coating, the surface of the coating cracks and initiates cracks, and under the action of stress, it expands to the interior of the coating (Y-direction) until the interface between the coating and the substrate. Finally, the cracks expand laterally (X-direction) along the interface, resulting in the separation of the coating and the substrate and the damage of the coating. Dynamic observation of the simulation process shows that the crack damage of the coating goes through the following three stages.

The first stage: Cracks sprouted on the coating surface. Figure 3 shows the applied cloud diagram when cracks start to develop on the surface of the CrN-3Pt coating. The analysis shows that the indenter slip generates a maximum stress of 1.518 GPa on the coating surface, which is much higher than the yield strength of the coating of 0.93 GPa, causing the coating surface to crack and sprout cracks that extend along the Y-direction inclination. It was observed that the crack sprouting in both CrN and CrN-Pt coatings was consistent with that in CrN-3Pt coating.

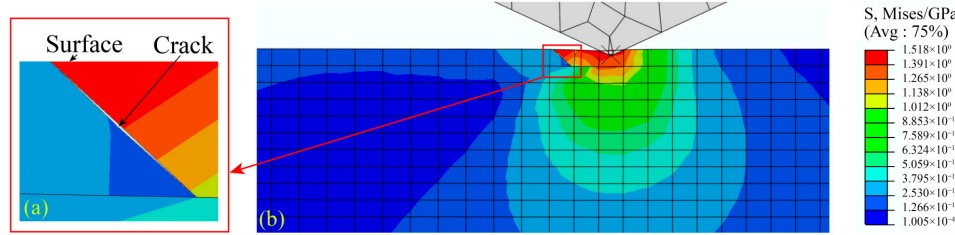

**Figure 3.** Application cloud diagram of CrN-3Pt coating surface when cracks start to appear: (**a**) Crack enlargement, (**b**) State of crack.

In order to analyze the relationship between the three stress components S11, S22 and S12 and crack sprouting, the distribution values of the three stress components along the X-direction of the coating surface when cracks start to occur in the three coatings were extracted separately (as shown in Figure 4), and the stress gradients of the three stress components were calculated as shown in Figure 5. When the coating surface cracks, the first large abrupt change in stress in all directions of the surface will occur at the crack location [31,32].

Therefore, according to Figure 4a–c, it can be measured that the surface of the CrN coating sprouts cracks at X = 21.61 μm, the surface of the CrN-Pt coating sprouts cracks at X = 21.36 μm, and the surface of the CrN-3Pt coating sprouts cracks at X = 22.06 μm. Further measurement of the indenter position at this point shows that the indenter of all three coatings is at the position where the S22 trough is located, which indicates that the S22 stress component is mainly generated by the normal pressure of the indenter. According to the crack location, indenter location, dynamic observation of the simulation process, and comprehensive analysis of Figures 4 and 5, the mechanism of the three coatings sprouting can be seen as follows.

CrN coating: Cracks were generated behind the indenter (left side), S11 and S22 dominated the crack sprouting, S12 synergized, and the cracks were tilted and distributed. The three-way stress synergized to first cause the coating to develop superficial localized tilted delamination cracking along the X-direction, and then it extended to the coating surface and interior.

CrN-Pt and CrN-3Pt coatings: Cracks were generated behind the indenter (left side), S11 dominated the crack sprouting, S22 and S12 cooperated and the cracks were distributed tilted. The three-way stresses cooperated to crack the coating surface along the Y-direction tilt and then extend to the inside of the coating.

The combined results show that the dynamic friction load on the coating surface is the main reason for the initiation of cracks on the three coating surfaces. In addition, the comparative analysis of various stress gradients of the CrN, CrN-Pt and CrN-3Pt coatings shows that the three-dimensional stress gradient of CrN is the smallest. With the increase in Pt element content, the three-dimensional stress gradient of CrN-Pt and CrN-3Pt coatings gradually increases, indicating that the fracture resistance of the CrN coating is the best, and the fracture resistance of the CrN-Pt and CrN-3Pt coatings decreases in turn.

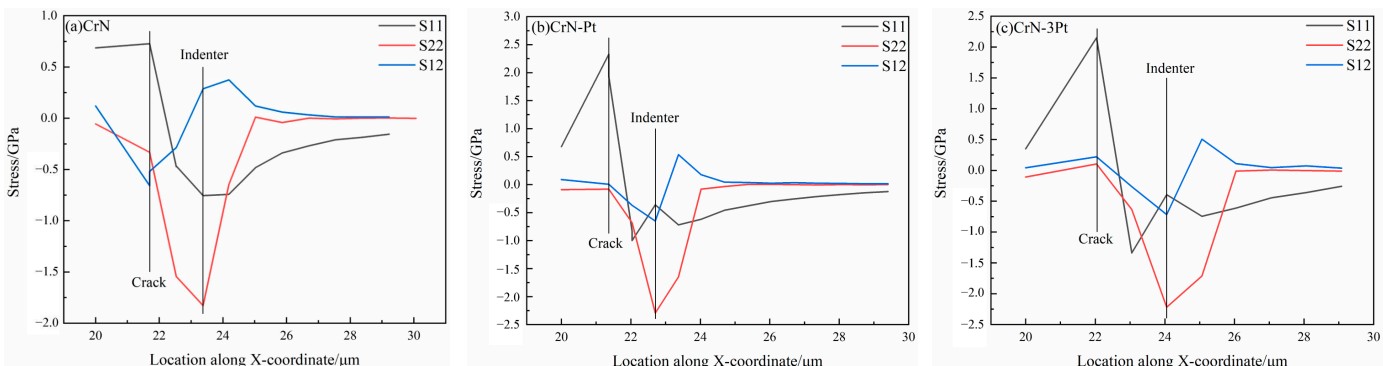

**Figure 4.** Isotropic stress distribution along the X-direction on the surface of three coatings. (**a**) CrN; (**b**) CrN-Pt; (**c**) CrN-3Pt.

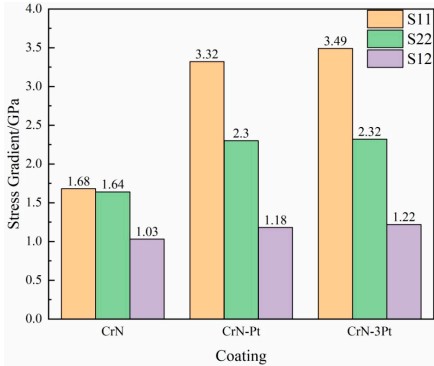

**Figure 5.** Isotropic stress gradients on the surfaces of three coatings.

The second stage: Downward expansion of coating cracks. Figure 6 shows the stress clouds of the CrN-3Pt coating when the cracks extend to half of the coating thickness. Analyzing Figure 6 and observing the dynamic simulation process of crack extension of the three coatings, it is clear that as the indenter continues to slip, stresses continue to be generated inside the three coatings. Under the stress, the cracks expand along the Y-direction toward the deeper part of the coating. At the same time, many irregularly arranged small sub-cracks are generated along the X-direction on the side of the main crack, so that the coating presents a synergistic damage state with cracking mainly along the Y-direction and delamination along the X-direction.

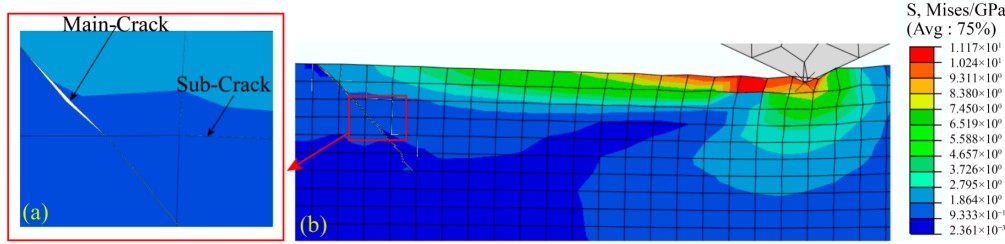

**Figure 6.** Stress nephogram of the CrN-3Pt coating when the crack propagates to half of the coating thickness: (**a**) Crack enlargement, (**b**) State of crack.

In order to analyze the relationship between coating extension and three-way stress in detail, two sets of stress component values were extracted from the moment when the main crack extended to half of the coating thickness: one is the distribution of three-way stress along the X-direction in the middle of the crack at this moment (Figure 7), and the other is the distribution of three-way stress along the X-direction at the tip of the crack at this moment (Figure 8). At the same time, the three-direction stress gradient at this moment is calculated (Figure 9a,b).

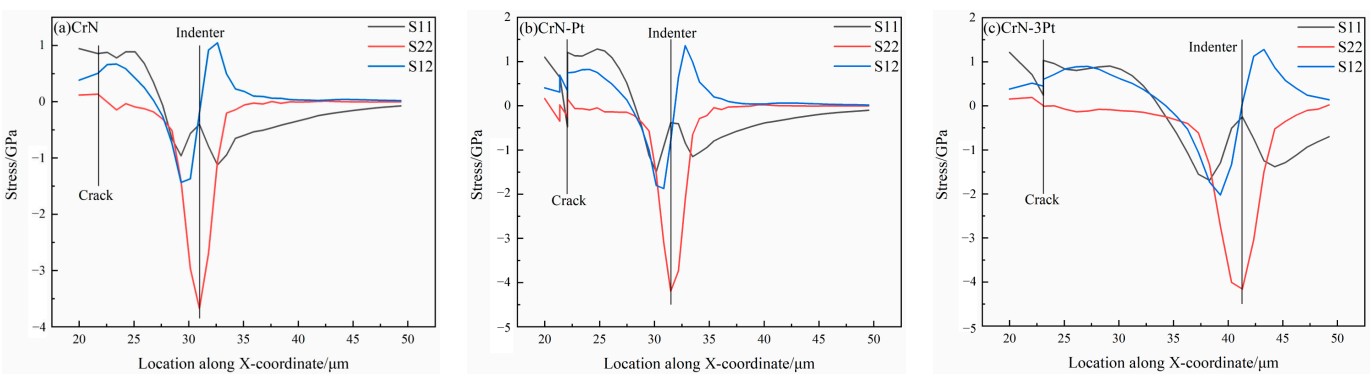

**Figure 7.** Anisotropic stress distribution in the middle of the three coating cracks when the crack propagates to half of the coating thickness. (**a**) CrN; (**b**) CrN-Pt; (**c**) CrN-3Pt.

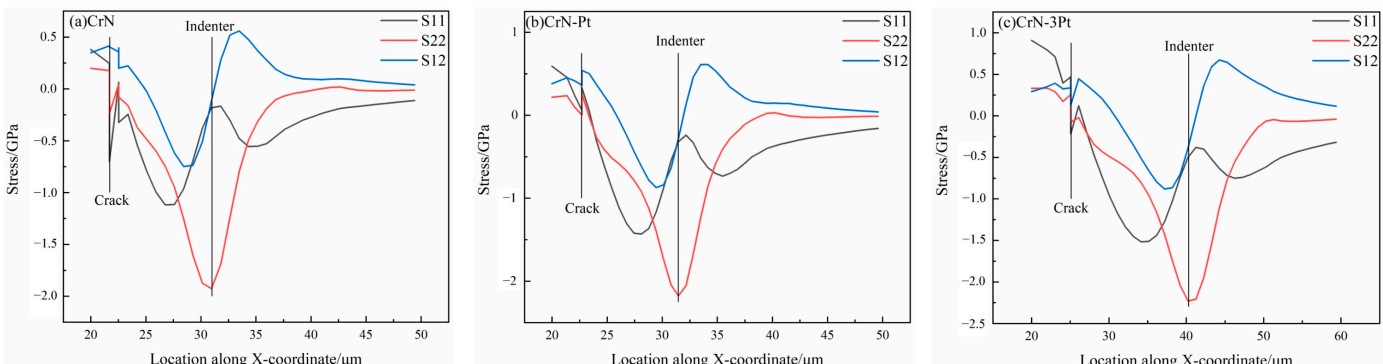

**Figure 8.** Anisotropic stress distribution at the crack tip of three coatings when the crack propagates to half of the coating thickness. (**a**) CrN; (**b**) CrN-Pt; (**c**) CrN-3Pt.

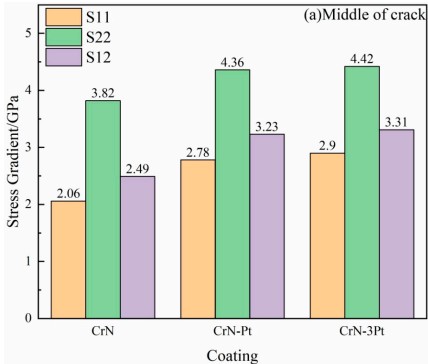
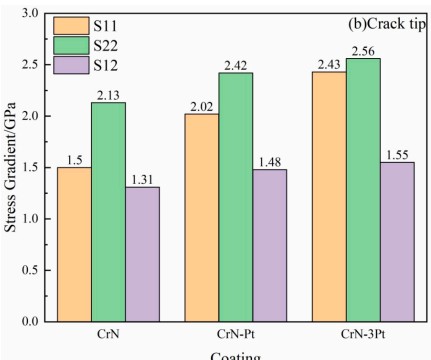

**Figure 9.** Isotropic stress gradients when cracks propagate to half the thickness of the three coatings. (**a**) Middle of crack; (**b**) Crack tip.

Analyzing Figures 7–9 and observing the dynamic simulation process of the three coating cracks expansion, it is clear that the crack is at the first sudden change of the three-way stress at this moment, the indenter is at the maximum valley of S22, and the indenter is in front of the crack. S22 dominates the scale expansion of the existing main crack, and S12 dominates the generation of a small Y-directional sub-crack next to the main crack. S22 also dominates the expansion of the main crack to the coating depth, which is caused by the equivalent auxiliary effect of S11 and S12, resulting in an inclined expansion of the main crack (oblique to the side of the indenter). Therefore, the extrusion load on the coating surface is the main cause of the scale expansion and downward expansion of the three coating cracks.

The third stage: Separation of coating and substrate. Figure 10 is the stress nephogram of CrN-3Pt coating when the crack extends to the interface between the coating and the substrate. Analyzing Figure 10 and observing the dynamic simulation process of crack propagation of the three coatings, it can be seen that when the main crack extends to the interface between the coating and the substrate, it will expand laterally along the X-direction, causing the separation of the coating and the substrate, making the coating close to the state of falling off damage.

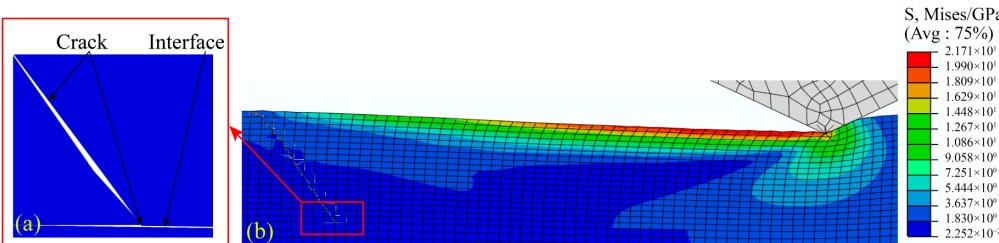

**Figure 10.** Stress nephogram of the CrN-3Pt coating when the crack propagates to the coating-substrate interface: (**a**) Crack enlargement, (**b**) State of crack.

In order to analyze the relationship between the separation of coating and substrate and the three-way stress in detail, two sets of stress component values are extracted when the main crack extends to the interface between the coating and substrate: one is the distribution value of the three-way stress in the middle of the crack along the X-direction at this time (see Figure 11), and the other group is the distribution value of the three-way stress at the crack tip along the X-direction at this time (see Figure 12). At the same time, the three-direction stress gradient at this moment is calculated (Figure 13a,b).

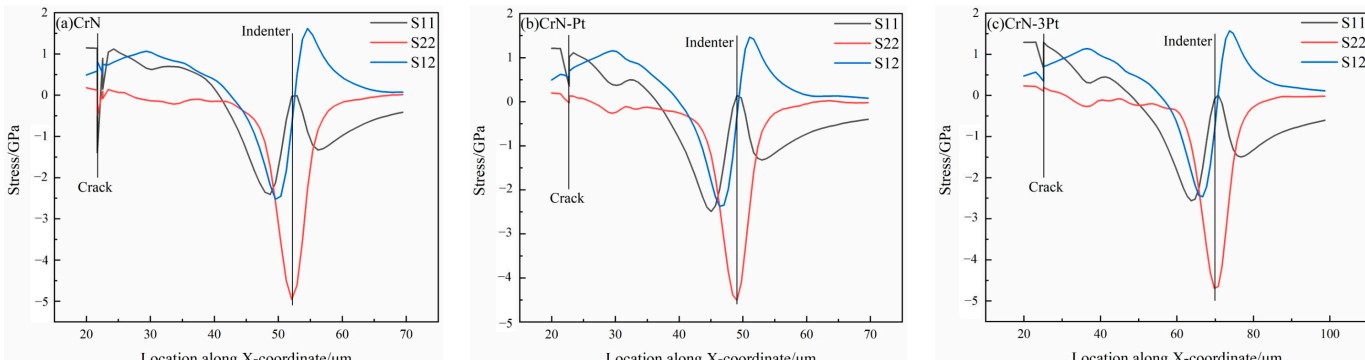

**Figure 11.** Anisotropic stress distribution in the middle of the three coating cracks when the crack propagates to the bonding surface of the film base. (**a**) CrN; (**b**) CrN-Pt; (**c**) CrN-3Pt.

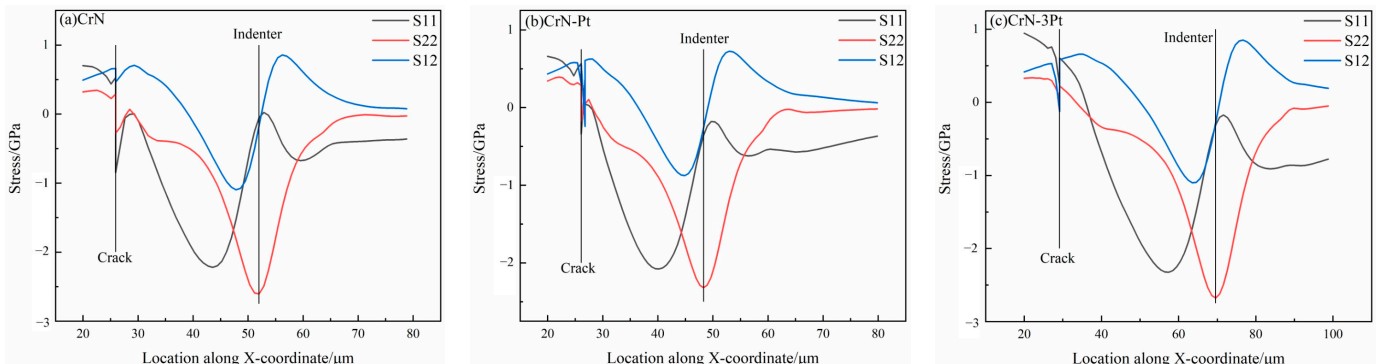

**Figure 12.** Anisotropic stress distribution at the crack tip of the three coating when the crack propagates to the bonding surface of the film base. (**a**) CrN; (**b**) CrN-Pt; (**c**) CrN-3Pt.

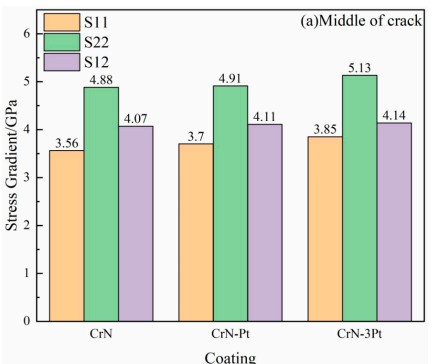 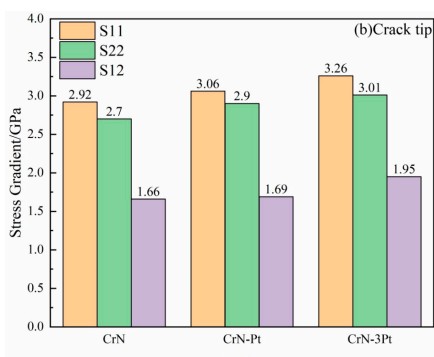

**Figure 13.** Anisotropic stress gradients of three coating cracks propagating to the coating–substrate interface. (**a**) Middle of crack; (**b**) Crack tip.

Analyzing Figures 11–13 and observing the dynamic simulation process of the three coating cracks expanding to the coating–substrate interface, it can be seen that at this moment, the crack is in the position of the first abrupt change of the three-way stress, the indenter is at the maximum valley of S22, and the indenter is in front of the crack. S11 and S22 dominated the lateral expansion of the main crack in the X-direction, causing the separation of the coating–substrate. Thus, the extrusion load on the coating surface in concert with the dynamic friction load caused the coating–substrate separation.

In order to study the influence of doping Pt elements with different contents on the fracture resistance behavior of the CrN coating surface, the distribution of three-dimensional residual stress components along the X-direction on the coating surface after the simulation

is extracted respectively, as shown in Figure 14, and the three-dimensional residual stress gradient is calculated as shown in Figure 15. Observation shows that the coating surface is dominated by the residual stress S11 tensile stress, with a large gradient, while S22 and S12 are almost zero with a small gradient, indicating that the S11 residual tensile stress of the three coating surfaces will become the main reason for the coating surface to resist fracture behavior under loading. The maximum residual tensile stress S11 values of the three coatings in Figure 14 are extracted and compared (see Figure 16a): the maximum residual tensile stress S11 value of the CrN coating is the smallest, and its surface is the least prone to fracture. The maximum residual tensile stress S11 of the CrN-Pt coating increases, and the degree of surface fracture decreases. With the increase in Pt content, the maximum residual tensile stress S11 value of the CrN-3Pt coating reaches the maximum, and the surface is the most prone to fracture. The maximum residual tensile stress S11 gradient of the three coating surfaces is further compared and analyzed (see Figure 16b), which also proves the analysis results of the fracture resistance behavior of the three coating surfaces.

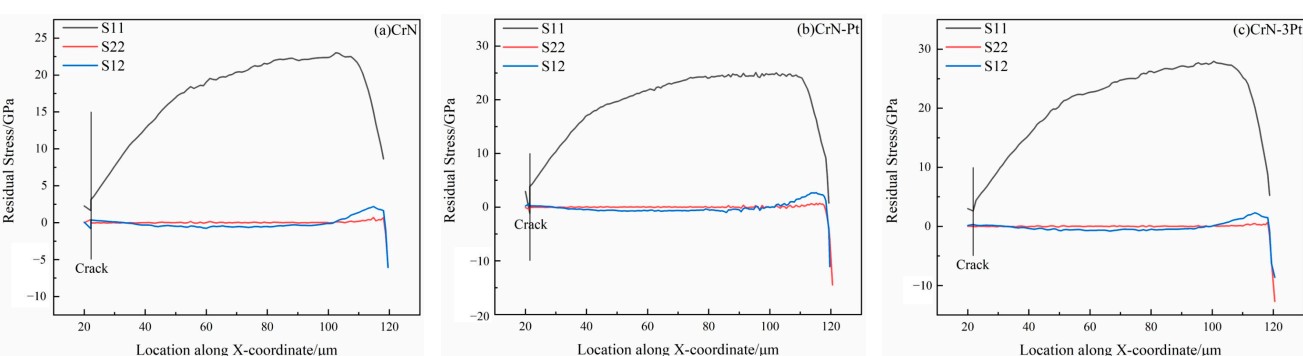

**Figure 14.** Isotropic residual stress distribution on the surfaces of three coatings. (**a**) CrN; (**b**) CrN-Pt; (**c**) CrN-3Pt.

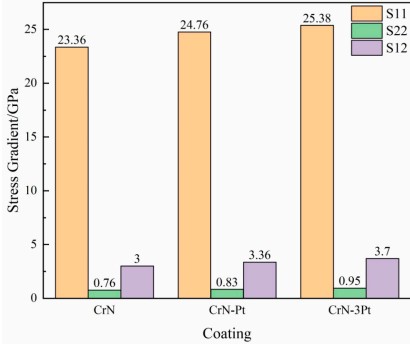

**Figure 15.** Isotropic residual stress gradients on the surfaces of three coatings.

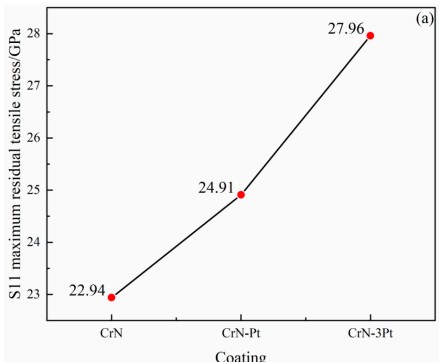
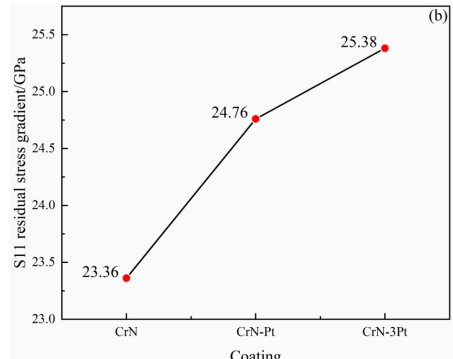

**Figure 16.** Comparison of the maximum value and gradient of the maximum residual tensile stress S11 on the surfaces of the three coatings. (**a**) S11 residual stress; (**b**) S11 residual stress gradient.

### 3.2. Fracture Analysis of Coating and Substrate

When the crack is fully extended to the coating–substrate interface, the critical loads of the three coatings are measured and obtained as shown in Table 4. The adhesion energy between the coating and substrate of the three coatings can be calculated according to Equation (1) as shown in Table 4. The comparison analysis shows that the film base bond strength of the CrN coating is the highest, the film base bond strength of the CrN-Pt coating decreases, and the film base bond strength of the CrN-3Pt coating is the worst, which indicates that the CrN coating is more difficult to peel off. The distribution of the three-way residual stresses along the X-direction of the bonding surface of the three coatings is shown in Figure 17, and the gradient of the three-way residual stresses is calculated in Figure 18, which shows that the maximum residual stresses at the bonding surface of the three coatings are still dominated by the S11 tensile stresses and have the largest gradient, indicating that the S11 residual tensile stresses at the bonding surface of the three coatings will be the main cause of the separation of the coatings under loading. Comparing and analyzing the S11 gradient values of the three coatings, it can be seen that the S11 gradient value of the CrN coating is the smallest, the bonding strength of the coating–substrate is the best, while the S11 gradient of the CrN-3Pt coating is the largest, and the bonding strength of the coating–substrate is the worst, which is consistent with the calculation results in Table 4.

**Table 4.** Critical loads and film-substrate binding energies of the three coatings.

| Coating | CrN | CrN-Pt | CrN-3Pt |
|---|---|---|---|
| Critical load (mN) | 25.58 | 26.83 | 44.06 |
| Bonding energy (J/m$^2$) | 0.50 | 0.45 | 0.41 |

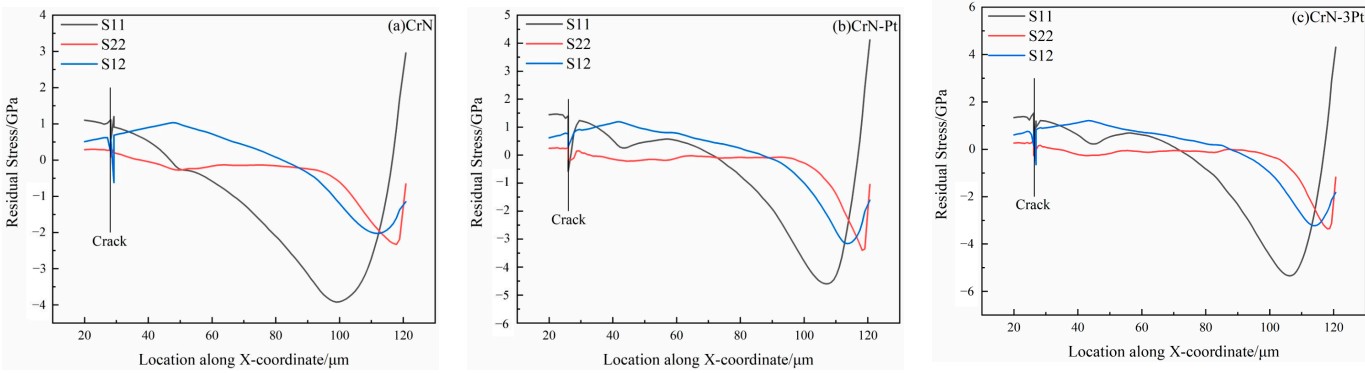

**Figure 17.** Distribution of residual stress at the interface between three coatings and substrate. (**a**) CrN; (**b**) CrN-Pt; (**c**) CrN-3Pt.

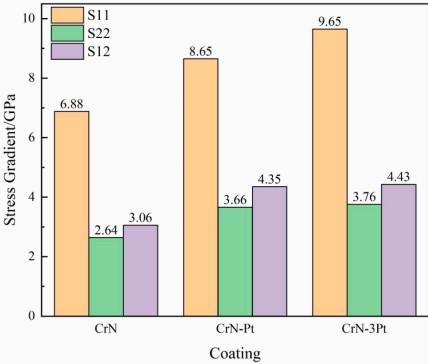

**Figure 18.** Isotropic residual stress gradient at the interface between the three coatings and substrate.

*3.3. Analysis of Scratch Test*

After the scratch test, the SEM images of the surface morphology of CrN, CrN-Pt and CrN-3Pt coatings are shown in Figure 19. It can be seen from the observation that under the same indenter load, cracks along the Y-direction (coating depth direction) have been generated in the three coatings, but different crack morphology has been formed due to the different wear resistance of the coatings: CrN coating cracks are small and have a streak-like regular arrangement, and the cracks are not cross-through, indicating that the crack damage is light. The scale of CrN-Pt cracks increases and there is cross-through between the cracks, which indicates that the cracks are generated and there is transverse and longitudinal expansion, and the degree of crack damage increases. CrN-3Pt cracks have the largest scale and disorderly arrangement, with serious transverse and longitudinal expansion and cross-through between cracks, indicating the heaviest degree of crack damage. The above results show that among the three coatings, the CrN coating has the best fracture resistance, the CrN-Pt coating has reduced fracture resistance, and the CrN-3Pt coating has the worst fracture resistance. Therefore, doping Pt in the CrN coating will reduce the fracture resistance of the coating, and the more Pt element content, the worse the fracture resistance of the coating and the more serious the damage. This result is consistent with the simulation result, which verifies the correctness of the simulation result.

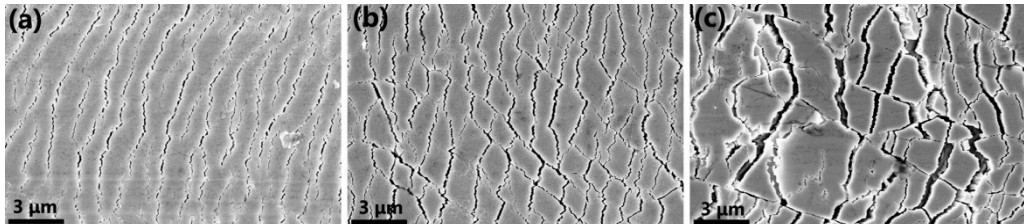

**Figure 19.** SEM image of the microstructure of the coatings scratches: (**a**) CrN, (**b**) CrN-Pt, (**c**) CrN-3Pt.

## 4. Conclusions

(1) The protective coating on the surface of electrical connectors bears the dynamic friction load and extrusion load on the surface during the plugging process. The fracture behavior of the coating under the load mainly includes three stages: crack initiation, propagation and separation between the coating and the substrate. S11 and S22 jointly guide the transverse crack initiation on the surface of CrN coating, and S11 guides the longitudinal crack initiation on the surface of the CrN-Pt coating and the CrN-3Pt coating first. The main crack of S22, which dominates the initiation of the three coatings, continues to expand obliquely to the interior of the coating. S11 and S22 jointly led the main crack to continue to expand along the film base interface, resulting in the separation of the coating and the substrate.

(2) The simulation and test results show that the CrN coating without the Pt element is the most difficult to fracture. The fracture degree of the CrN-Pt coating decreases, and it is relatively easy to fracture. With the increase in Pt content, the CrN-3Pt coating is the easiest to fracture. It shows that the fracture degree of the CrN coating decreases with the increase in doping Pt element content.

(3) The simulation and calculation results show that the bonding energy of the film base of the CrN coating without the Pt element is 0.50 J/m$^2$, and the bonding strength between the coating and the substrate is the best. The film base bonding energy of the CrN-Pt coating is 0.45 J/m$^2$, and the bonding strength between the coating and the substrate decreased. With the increase in the Pt content, the bonding energy of the film substrate of the CrN-3Pt coating was 0.41 J/m$^2$, and the bonding strength between the coating and the substrate was the worst. It shows that the bonding strength of the CrN coating and substrate decreases with the increase in the content of doped Pt element.

**Author Contributions:** Methodology, writing—original draft preparation, software, H.S.; validation, W.Z.; writing—review and editing, supervision, formal analysis, Y.F.; data curation, visualization, S.H.; conceptualization, funding acquisition, H.T.; project administration, resources, Z.X.; All authors have read and agreed to the published version of the manuscript.

**Funding:** This study was supported by the National Natural Science Foundation of China (52171076), Science and Technology Research of Liaoning Province Education Department (JYT19035).

**Institutional Review Board Statement:** Not applicable.

**Informed Consent Statement:** Not applicable.

**Data Availability Statement:** No new data were created or analyzed in this study. Data sharing is not applicable to this article.

**Conflicts of Interest:** The authors declare no conflict of interest.

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
