# Peer review of "Exploring the Effect of Pt Addition on the Fracture Behavior of CrN Coatings by Finite Element Simulation"

_coatings, doi:10.3390/coatings12081131_

Round 1

Reviewer 1 Report

This is an interesting study. The authors mainly use simulations, but experiments have also been carried out. The described work has an obvious applied orientation, although it also has some scientific value. I think that this work will find readers and therefore may be published after some improvements.
1. The paper does not explain why CrN-3Pt is one and a half times thicker than other coatings. This needs to be explained and whether this increase has affected the results of a comparative study of the three coatings.
2. In paragraph "2.1 Finite element simulation mode" I would suggest combining tables 2 and 3.
3. I think the English language of the paper should be improved somewhat. Phrases like "Take Φ 25mm × 8mm cylinder 316L stainless steel substrate..." sound too imperative.
4. The autors should be more careful about the design of the text. For example, the link to figure 9 in the text is before the links to figures 7 and 8, the link to 13 is earlier than to 12.

Author Response

感谢您阅读我的文章。请参阅附件。

Reviewer 2 Report

The authors have studied the effect of Pt addition on the fracture behavior of CrN coatings. This work is interesting for manufacturers of electrical connectors but there are some issues that need to be addressed before publication.

Abstract: "S11 and S22 caused transverse cracks [...]". What do "S11 and S12" mean? The abstract should be written as a stand-alone document.

Introduction section is to general.  What results and under what conditions were found in papers [1-4], [5-8], [10-13], [14-15]? Group quotation makes it difficult to understand what specifically has been found.

First paragraph from the 2.1. section should be presented as separate subsection "Material".

lines 80-81: "Through the preliminary experimental and simulation studies, we obtained the elastic-plastic parameters of [...]." The details of preliminary experimental and simulation studies should be included in the manuscript. This would allow others to replicate and build on published results.

The "Mesh division" subsection: The authors did not perform mesh sensitivity analysis. Results of mesh sensitivity analysis were not presented in this manuscript.

Table 4: It is impossible to run elastic-plastic analysis without "Poisson's ratio". So, the results are debatable.

The authors in elastic-plastic material model did not take into account the work-hardening effect. Establishing elastic / perfectly-plastic model seems too simplistic.

The contact conditions have not been clarified. What friction model was used? How was the value of the coefficient of friction determined? Its value was also not disclosed. In contact analysis friction conditions play a key role.

lines 53-54: The authors say that "At the same time, the results of scratch test and finite element simulation are compared [...]" However, the details of the scratch test procedure have not been presented in the "Material and Methods" section. Subsection 3.3 should only contain the results.

It is not known what material parameters and what kind of finite elements were assigned to the indenter geometry.

Units of stresses are not specified in Figure 2, 3, … etc.

line 123: "equivalent force Mises" ???

Figure 4. The font size is too small. It is suggested to enlarge the size of the drawings. This also applies to figures 7, 8, 11, 12, 14, 17. The font size is also too small in figures 5, 9, 13, 16, 18.

lines 185-188. Figures should be called up in the text in the order in which they appear in the article. After drawing 6 is called, Figure 9 cannot be cited immediately.

In the Conclusions, please add the most important quantitative results.

The "References" section is not formatted according to the "Instructions for authors".

Data Availability Statement: In this section, please provide details regarding where data supporting reported results can be found, including links to publicly archived datasets analyzed or generated during the study. Please refer to suggested Data Availability Statements in section “MDPI Research Data Policies”. You might choose to exclude this statement if the study did not report any data. This paper reports data... so you can not state "Not applicable".

Author Response

Thanks for reviewing my article.Please see the attachment.

Reviewer 3 Report

In the manuscript the authors presented an interesting work devoted to the finite element simulation method to explore the influence of different content of pltinum on the fracture behavior of CrNPt coating. The authors compared obtained results and the results of scratch tests to verify the accuracy of the used finite element method. The results can provides support for the design and preparation of CrN protective coating for electrical connectors.

The topic of the manuscript is valuable for science and practice. This work may be of interest to scientific community and I believe the paper is worth publishing. 

The manuscript is well-organized and readable, the work is sufficiently detailed and scientifically documented.

The manuscript requires minor revision. My specific comments and suggestions:

1) The designations of alloys adopted by the authors (CrN, CrN-Pt, CrN-3Pt) may be misinterpreted by the reader of the publication regarding the amount of the Pt addition. I suggest to change these designations so that they are directly related to the amount of platinum, i.e. Cr-5Pt, Cr-15Pt. Then, when reading the paper, you will immediately know what amount of platinum you are dealing with.

2) section Results: What is equivalent force Mises? Please explain shortly what the von Mises criterion states.

3)  lines 307-310: This paragraph which describes the apparatus and parameters of the scratch test should be moved to the section Materials and Methods.

4) References: Following the editor's instructions for authors, journal references must cite also digital object identifier (DOI) where available.

Author Response

(The authors gave the same response as above.)

Round 2

Reviewer 2 Report

I am satisfied with the reviewed version and the provided explanations to all raised questions. I recommend the acceptance of the manuscript.

Author Response

We thank the reviewer for reading our paper carefully and agreeing to accept the paper.